# AdsGT: Graph Transformer for Predicting Global Minimum Adsorption Energy

## Abstract

The fast assessment of the binding strength between adsorbates and catalyst surfaces is crucial for catalyst design, where global minimum adsorption energy (GMAE) is one of the most representative descriptors. However, catalyst surfaces typically have multiple adsorption sites and numerous possible adsorption configurations, which makes it prohibitively expensive to calculate the GMAE using Density Functional Theory (DFT). Additionally, most machine learning methods can only predict local minimum adsorption energies and rely on information about adsorption configurations. To overcome these challenges, we designed a graph transformer (AdsGT) that can predict the GMAE based on surface graphs and adsorbate feature vectors without any binding structure information. To evaluate the performance of AdsGT, three new datasets on GMAE were constructed from OC20-Dense, Catalysis Hub, and FG-dataset. For a wide range of combinations of catalyst surfaces and adsorbates, AdsGT achieves test mean absolute errors of 0.10 and 0.14 eV on the two GMAE datasets respectively, demonstrating its good reliability and generalizability.

## 1 Introduction

The adsorption energy of an adsorbate on the catalyst surface is crucial for determining the reactivity and selectivity of catalytic reactions. The highest catalytic activity of a material will be achieved at the optimal adsorption energy for a specific reaction, according to the Sabatier principle [1, 2] (Fig. 1). Therefore, developing cheap and efficient adsorption energy evaluation methods are of great significance for catalyst discovery. Currently, high-throughput screening of catalysts relies heavily on computationally expensive simulations like Density Functional Theory (DFT) [3, 4]. However, multiple adsorption sites and variable adsorbate geometries lead to numerous possible adsorption configurations and local minima on the binding energy surface [5, 6]. The local adsorption energy strongly depends on the initial structure of the simulation and cannot provide a fair evaluation of different catalysts. Several methods, including global optimization algorithms [7–9] and "brute-force" searches [10, 11], have been employed to find the most stable adsorption structures and corresponding global minimum adsorption energies (GMAE). Unfortunately, the exponential rise in computational costs renders these methods inadequate for the screening of diverse catalyst candidates.

Machine learning (ML) holds the potential to approximate DFT-level accuracy at significantly lower time costs [12, 13]. A lot of ML models, such as random forests, multilayer perceptions, and graph neural networks, have been explored to predict adsorption energy of adsorbate-surface systems [14–17]. However, several drawbacks are present in most models, which (1) can only predict local

Submitted to NeurIPS 2023 AI for Science Workshop.

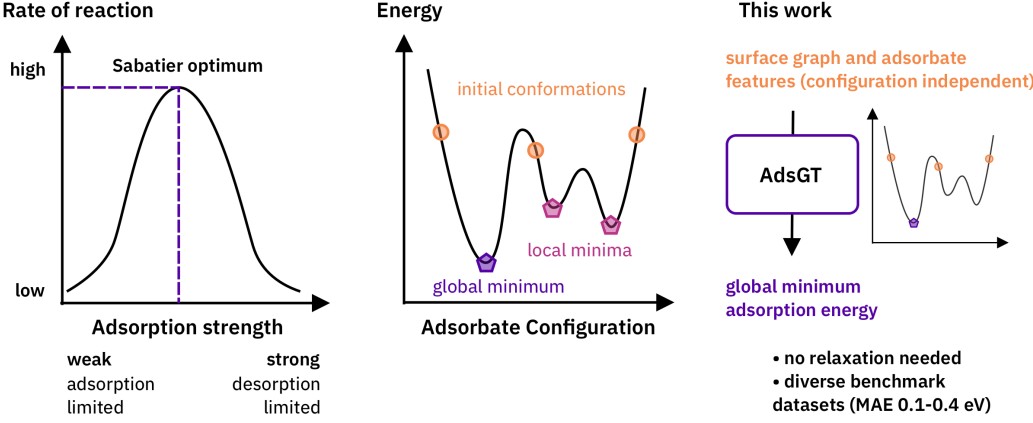

Figure 1: **Overview** Left: The Sabatier principle describes that a catalyst should bind a substrate neither too weakly nor too strongly. Middle: Global and local minima adsorbate configurations on the catalytic surface. Right: The global minimum adsorption energy prediction task is addressed in this work without requiring adsorption configuration information.

minimum adsorption energies, (2) require binding information between the adsorbates and catalyst surfaces, and (3) exhibit a poor generalizability limited to specific adsorbates. Recently, Ulissi et al. proposed the AdsorbML workflow [18], which combines heuristic search and ML potentials to accelerate the GMAE calculation. The ML potentials trained on the huge Open Catalyst (OC)20 dataset achieve promising prediction accuracy and substantial speedups over DFT computations [18]. Moreover, Margraf et al. [5] proposed a global optimization protocol that employs on-the-fly ML potentials trained on iteratively DFT calculations to search the most stable adsorption structures. This method is versatile for various combinations of surfaces and adsorbates, and significantly reduces the reliance on prior expertise and the number of required DFT calculations [5].

Herein, a new strategy for directly predicting GMAE without binding structure information is proposed. A novel graph transformer model, called AdsGT, was designed for the GMAE prediction based on the surface graphs and adsorbate feature vectors. Three datasets on GMAE were constructed and applied for model evaluation. AdsGT demonstrates excellent performance in predicting GMAE, with mean absolute errors (MAE) below 0.14 eV for two of the datasets and 0.51 eV on a more challenging dataset with fewer data points. A pretraining strategy was also proposed to improve AdsGT performance to a MAE of 0.43 eV. All results highlight the learning ability of AdsGT for catalytic surface chemistry and its association with adsorbates. This work makes a valuable contribution to accelerating GMAE calculations and catalyst screening.

## 2 Methods

### 2.1 Datasets

The datasets for the global minimum adsorption energies in this study come from OC20-Dense [18], Catalysis Hub [19], and 'functional groups' (FG)-dataset [6] datasets. Each of the source datasets enumerated all adsorption sites on surfaces and performed DFT calculations on various possible adsorption configurations. The data cleaning was conducted to take the lowest adsorption energy of all conformations for each combination of catalyst surface and adsorbate as the global minimum adsorption energy. Subsequently, three new datasets, named OCD-GMAE, Alloy-GMAE and FG-GMAE, were constructed, and each data point represents a unique combination of catalyst surface and adsorbate (Table 1). Random splitting is adopted on three datasets during the model training. More challenging splits will be investigated in future work.

Table 1: Overview of three new datasets on GMAE. () values represent the numbers of element types.

| Dataset | Combination Num. | Surface Num. | Adsorbate Num. | Range of GMAE (eV) |
|---|---|---|---|---|
| OCD-GMAE | 973 | 967 (54) | 74 (4) | -8.0 ∼ 6.4 |
| Alloy-GMAE | 11,260 | 1,916 (37) | 12 (5) | -4.3 ∼ 9.1 |
| FG-GMAE | 3,308 | 14 (14) | 202 (5) | -4.0 ∼ 0.8 |

In addition, a similar data cleaning procedure was employed on the OC20 dataset [20] to create a new dataset named OC20-LMAE, which comprises surface/adsorbate pairings along with their local minimum adsorption energies (LMAE). The OC20-LMAE dataset contains 345,254 data points and serves as an effective resource for model pretraining.

## 2.2 Surface graph

Each input catalyst surface is modeled as a graph $\mathcal{G}$ consisting of $n$ nodes (atoms) $\mathcal{V} = \{v_1, \ldots, v_n\}$ and $m$ edges (interactions) $\mathcal{E} = \{\epsilon_1, \ldots, \epsilon_m\} \subseteq \mathcal{V}^2$. $\mathbf{H} = [\boldsymbol{h}_1, \boldsymbol{h}_2, \cdots, \boldsymbol{h}_n]^T \in \mathbb{R}^{n \times k}$ is the node feature matrix, where $\boldsymbol{h}_i \in \mathbb{R}^k$ is the $k$-dimensional feature vector of atom $i$. $\mathbf{E} \in \mathbb{R}^{m \times k'}$ is the edge feature matrix, where $\boldsymbol{e}_{ij}^t \in \mathbb{R}^{k'}$ is the $k'$-dimensional feature vector of $t$-th edge between node $i$ and $j$. $\mathbf{X} = [\boldsymbol{x}_1, \boldsymbol{x}_2, \cdots, \boldsymbol{x}_n]^T \in \mathbb{R}^{n \times 3}$ is the position matrix, where $\boldsymbol{x}_i \in \mathbb{R}^3$ is the 3D Cartesian coordinate of atom $i$. For periodic boundary conditions (PBC), let the matrix $\mathbf{C} = [\boldsymbol{a}, \boldsymbol{b}, \boldsymbol{c}]^T \in \mathbb{R}^{3 \times 3}$ depicts how the unit cell is replicated in three directions $\boldsymbol{a}$, $\boldsymbol{b}$ and $\boldsymbol{c}$.

**Periodic invariance** Ignoring periodic invariance will lead to different surface graphs and energy predictions for the same surface [21]. Different from crystals, the presence of the vacuum layer breaks the periodicity along the direction perpendicular to the surface. This means that the catalyst surfaces actually exhibit periodicity only in the $\boldsymbol{a}$ and $\boldsymbol{b}$ directions. Thus, the infinite surface structure can be represented as

$$\hat{\mathbf{H}} = \left\{ \hat{\boldsymbol{h}}_i \mid \hat{\boldsymbol{h}}_i = \boldsymbol{h}_i, \ i \in \mathbb{Z}, 1 \le i \le n \right\},$$
$$\hat{\mathbf{X}} = \left\{ \hat{\boldsymbol{x}}_i \mid \hat{\boldsymbol{x}}_i = \boldsymbol{x}_i + k_1 \boldsymbol{a} + k_2 \boldsymbol{b}, \ i, k_1, k_2 \in \mathbb{Z}, 1 \le i \le n \right\}. \quad (1)$$

To encode such periodic patterns, the infinite representation of the surface is used for graph construction, and all nodes and their repeated duplicates are considered to build edges. Given a cutoff radius $r_c \in \mathbb{R}$, if there is any integer pair $(k_1', k_2')$, such that the Euclidean distance $d_{ji} = \|\boldsymbol{x}_j + k_1'\boldsymbol{a} + k_2'\boldsymbol{b} - \boldsymbol{x}_i\|_2 \le r_c$, then an edge is constructed from $j$ to $i$ with the initial edge feature $d_{ji}$. It should be pointed out that self-loop edges ($i = j$) are also considered if there exists any integer pair $(k_1', k_2')$ other than $(0, 0)$ such that $d = \|k_1'\boldsymbol{a} + k_2'\boldsymbol{b}\|_2 \le r_c$.

**Positional feature** Unlike molecular graphs, the importance of each atom in the catalyst surface is different for adsorption energy prediction (Fig. 2). For example, atoms closer to the adsorbate are more important, while atoms at the bottom are less important. Moreover, GNNs cannot determine whether the atoms are located at the interface in contact with the adsorbate based on the surface graph. They cannot distinguish between interfacial atoms and subsurface atoms. To help models understand the varying importance of different atoms, each atom $i$ of the surface graph will get a positional feature $\delta_i$ computed by

$$\delta_i = \frac{h - h_{min}}{h_{max} - h_{min}} \quad (2)$$

where $h$ is the height of the atom $i$ and calculated by the projection length of the atom coordinate $\boldsymbol{x}_i$ on the $\boldsymbol{c}$ vector. $h_{max}$ and $h_{min}$ represent the maximum and minimum heights of surface atoms, respectively.

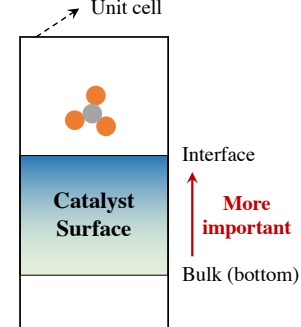

Figure 2: Illustration of the varying importance of different atoms on a catalyst surface.

## 2.3 Adsorbate feature

The representation of adsorbate is crucial for models to predict the lowest adsorption energy for a given combination of surface and adsorbate. Many adsorbate species, especially in the field of electrocatalysis, consist of fewer than five atoms. Some adsorbates, such as *H, *O and *NH have only one or two atoms. Therefore, molecular descriptors are used to represent adsorbates rather than the widely used molecular graphs. $\mathbf{P} = [\boldsymbol{p}_1, \boldsymbol{p}_2, \cdots, \boldsymbol{p}_s]^T \in \mathbb{R}^{s \times k''}$ is the adsorbate feature matrix, where $\boldsymbol{p}_c \in \mathbb{R}^{k''}$ is the $k''$-dimensional feature vector of the adsorbate for the surface/adsorbate combination $c$ ($1 \leq c \leq s$).

## 2.4 Model

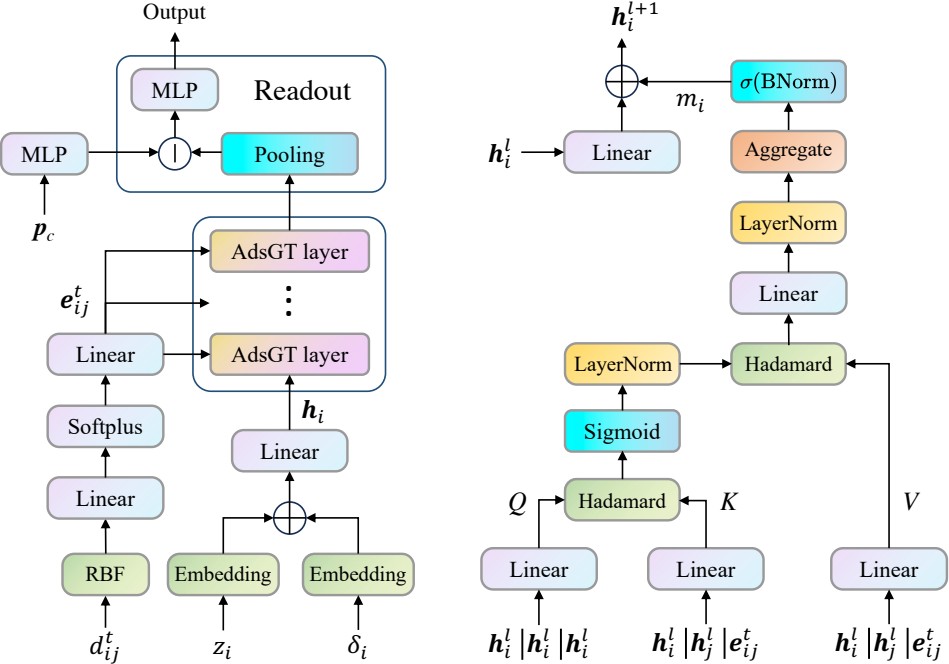

Figure 3: **Model architecture** of AdsGT (left) and its attention layer (right). $+$ and $|$ denote sum and concatenation operations, respectively. $\sigma$ denotes the activation function, and $\mathrm{BNorm}$ represents batch normalization.

The proposed AdsGT model (Fig. 3) consists of three parts: a graph encoder $E_G$, a vector encoder $E_V$, and a readout block $R_o$. Each surface/adsorbate combination $C$, consisting of a surface graph $\mathcal{G}_c = (\mathbf{H}, \mathbf{E})$ and an adsorbate feature vector $p_c$, is defined as the model input and the global minimum adsorption energy of the combination is set as the prediction target. A surface graph and an adsorbate feature vector are passed to the graph encoder $E_G$ and the vector encoder $E_V$ for embedding learning, respectively. Then, both embeddings are concatenated and passed to the readout block $R_o$ for the prediction of global minimum adsorption energy. The details of these parts are as follows.

**Graph encoder** In the initialization of $E_G$, atomic number $z_i$ and positional feature $\delta_i$ of node $i$ are passed to embedding layers to compute the initial node embedding $\boldsymbol{h}_i^0$. The distance $d_{ij}^t$ of $t$-th edge between node $i$ and $j$ is expanded via a set of exponential normal radial basis functions (RBF) and transformed by linear layers to obtain the edge embedding $\boldsymbol{e}_{ij}^t$. The message passing phase of $E_G$ follows the regular attention mechanism [21, 22]. In the $l$-th ($0 \leq l \leq L$) attention layer, edge-wise attention weights $\boldsymbol{\alpha}_{ij}^t$ and message $m_{ij}^t$ of $t$-th edge between node $i$ and $j$ are calculated based on $\boldsymbol{h}_i^l$, $\boldsymbol{h}_j^l$ and $\boldsymbol{e}_{ij}^t$ according to

$$\boldsymbol{q}_{ij} = W_Q^l \left( \boldsymbol{h}_i^l \Big| \boldsymbol{h}_i^l \Big| \boldsymbol{h}_i^l \right), \quad \boldsymbol{k}_{ij}^t = W_K^l \left( \boldsymbol{h}_i^l \Big| \boldsymbol{h}_j^l \Big| \boldsymbol{e}_{ij}^t \right), \quad \boldsymbol{v}_{ij}^t = W_V^l \left( \boldsymbol{h}_i^l \Big| \boldsymbol{h}_j^l \Big| \boldsymbol{e}_{ij}^t \right) \quad (3)$$

$$\boldsymbol{\alpha}_{ij}^t = \frac{\boldsymbol{q}_{ij} \circ \boldsymbol{k}_{ij}^t}{\sqrt{d_{\boldsymbol{k}_{ij}^h}}}, \quad \boldsymbol{m}_{ij}^t = \text{sigmoid}\left(\text{LNorm}\left(\boldsymbol{\alpha}_{ij}^t\right)\right) \circ \boldsymbol{v}_{ij}^t \tag{4}$$

where $W_Q^l$, $W_K^l$ and $W_V^l$ are three learnable weight matrices, $\circ$ represent the Hadamard product, and | denotes concatenation. LNorm denotes the layer normalization operation. Then, the message $m_i$ of node $i$ from all neighbors $\mathcal{N}_i$ is computed by

$$\boldsymbol{m}_i = \sum_{j \in \mathcal{N}_i} \sum_h \text{LNorm}\left(W_m^l \boldsymbol{m}_{ij}^t + b_m^l\right) \tag{5}$$

and the embedding of node $i$ is updated based on the message $m_i$ according to

$$\boldsymbol{h}_i^{l+1} = W_u^l \boldsymbol{h}_i^l + b_u^l + \sigma\left(\text{BNorm}\left(\boldsymbol{m}_i\right)\right) \tag{6}$$

where $W_m^l$ and $W_u^l$ are two learnable weight matrices, while $b_m^l$ and $b_u^l$ are two learnable bias vectors. $\sigma$ denotes the activation function, and BNorm represents batch normalization.

**Vector encoder**  A simple multilayer perceptron (MLP) is used to encode the feature vectors of adsorbates, and the adsorbate embedding of the combination $C$ is calculated based on

$$\boldsymbol{p}_c' = \text{MLP}(\boldsymbol{p}_c) \tag{7}$$

**Readout block**  For the surface/adsorbate combination $C$, graph-level embedding $\boldsymbol{g}_c$ of surface $\mathcal{G}_c$ is computed and concatenated with adsorbate embedding $\boldsymbol{p}_c'$ to predict the GMAE based on

$$\boldsymbol{g}_c = \sum_{i \in \mathcal{G}_c} \boldsymbol{h}_i^L, \quad y = \text{MLP}\left(\boldsymbol{g}_c \mid \boldsymbol{p}_c'\right) \tag{8}$$

# 3    Results and Discussion

Table 2: Test MAE and success rates of AdsGT on the three GMAE datasets. The success rate is the percentage of predicted GMAEs within 0.1 eV of the DFT-computed ground truth GMAEs. Energy MAE is also computed between predicted and ground-truth GMAEs. All results are from 5 replicate experiments with different random seeds.

|  | Alloy-GMAE (11,260) | FG-GMAE (3,308) | OCD-GMAE (973) | OCD-GMAE (Pretrained, 973) |
|---|---|---|---|---|
| Energy MAE (eV) $\downarrow$ | $0.1388 \pm 0.0072$ | $0.1053 \pm 0.0065$ | $0.5149 \pm 0.0545$ | $0.4296 \pm 0.0326$ |
| Success rate (%) $\uparrow$ | $67.25 \pm 1.11$ | $69.74 \pm 2.17$ | $13.47 \pm 4.85$ | $25.36 \pm 2.12$ |

The prediction performance of AdsGT was evaluated on the three GMAE datasets, and the results are depicted in Table 2. These three datasets have different characteristics: (1) Alloy-GMAE has a variety of surfaces (1916) but a small number of adsorbates (12), (2) FG-GMAE has a small number of surface types (14) but a large variety of adsorbates (202), and (3) OCD-GMAE contains a variety of surfaces (967) and adsorbates (74) but a smaller amount of data. As shown in the Table 2, AdsGT achieves excellent performance with MAE less than 0.14 eV and a success rate exceeding 67 % on the Alloy-GMAE and FG-GMAE datasets, without any binding structural information. However, AdsGT exhibits worse performance with an MAE higher than 0.5 eV on the OCD-GMAE, which comprises a broader range of surface/adsorbate combinations but fewer data points. Given the small-size constraint, AdsGT is pretrained on the larger dataset OC20-LMAE and finetuned on the OCD-GMAE. It results in a lower energy MAE (0.43 eV) and a higher success rate (25.4 %) compared to the directly training AdsGT. More work on transfer learning and data augmentation will be explored in the future.

Moreover, several models with the same AdsGT architecture but different graph encoders [23–26] are explored on the OCD-GMAE dataset (Table 3). The results indicate that our designed AdsGT graph encoder surpasses all baseline graph encoders, demonstrating its good learning capability in catalytic surface chemistry. Unfortunately, larger graph encoder from GemNet-OC model fails to achieve better performance on this small dataset with diverse surfaces and adsorbates.

Table 3: Test MAE of AdsGT and baseline models on the OCD-GMAE dataset. ∗ denotes replacing the graph encoder in the AdsGT architecture with the corresponding baseline graph encoder.

| Graph encoder | Energy MAE (eV) ↓ |
|---|---|
| *SchNet | $0.8743 \pm 0.0952$ |
| *CGCNN | $0.6832 \pm 0.0734$ |
| *DimeNet++ | $0.8839 \pm 0.0825$ |
| *GemNet-OC | $1.1437 \pm 0.0672$ |
| AdsGT | $\mathbf{0.5149 \pm 0.0545}$ |

## 4   Conclusion

Our work presents AdsGT, a novel graph transformer model for predicting global minimum adsorption energies of adsorbate-surface systems. AdsGT takes the combinations of surface graphs and adsorbate feature vectors as input without requiring any adsorption configuration information. On three datasets covering a wide range of surfaces and adsorbates, AdsGT demonstrates strong performance in predicting GMAE, with mean absolute errors within 0.14 eV for two of the datasets, and 0.43 eV on the more challenging dataset with fewer datapoints. The results highlight the ability of graph neural networks like AdsGT to learn meaningful representations of surface chemistry and approximate DFT adsorption energies. By rapidly predicting GMAE, AdsGT has the potential to accelerate high-throughput computational screening of novel catalysts. While AdsGT struggles on one dataset with greater diversity but fewer examples, transfer learning has been proved to be an effective measure to improve its generalizability. Overall, this work makes valuable contributions towards enabling graph ML models to guide the discovery of novel catalysts for renewable energy and industrial processes. The code and datasets will be publicly available to facilitate future research.

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
