# OpenReview forum: "AdsGT: Graph Transformer for Predicting Global Minimum Adsorption Energy"
_NeurIPS.cc/2023/Workshop/AI4Science — NeurIPS2023-AI4Science Poster_

### Official Review · Reviewer_6wuc · 2023-10-25
**An interesting paper for predicting global minimum adsorption energy to facilitate catalyst discovery**

**Rating:** 6
**Confidence:** 4

**Review:**

### Summary of the paper
This work studies the problem of predicting global minimum adsorption energy (GMAE) to facilitate catalyst discovery. Specifically, this paper first constructs three new benchmarking datasets for GMAE predictions, and then proposes AdsGT, a graph transformer for this task. AdsGT is evaluated on the three datasets and shows promising results.


### Strengths
- This paper studies an interesting and important application.
- Three benchmarking datasets are prepared, which may facilitate the study in this direction.
- AdsGT shows promising results and can work as a benchmark for future studies.
- A pre-training strategy is tried, showing potential in boosting performance when lack of labeled data, which can be especially useful for tasks in this domain.

### Weaknesses
- No baseline is included when benchmarking the results on the three datasets. Even though previous methods may follow different pipelines or require different info, it would still provide more info when including them as baselines.
- The used pre-training strategy is not described in the paper.
- Geometric equivariance is not well-explored in AdsGT. And the ablation study (table 3) may be less convincing since those GDL models are effective greatly due to their ability to capture geometric equivariance. It is unclear how these methods are adapted to this task, and thus it is hard to tell if the comparison is fair.

### Summary of the review
Overall, I think this is an interesting paper, and I appreciate the efforts put into preparing the benchmarking datasets. Yet, it would be better to explore more baselines and geometric equivariance in the study.

---

### Official Review · Reviewer_7gpk · 2023-10-25
**A novel application of graph transformer to the new task.**

**Rating:** 6
**Confidence:** 4

**Review:**

This paper demonstrate the effective power of graph transformer in the adsorption energy. They conduct experiments to validate this application. I think this is a novel application to a new task.

---

### Meta-Review · Area_Chair_R9Yb · 2023-10-27

**Recommendation:** Accept (Poster)
**Confidence:** 4

**Metareview:**

## Summary:
This work targets an emerging area in the field of catalysis screening and proposes novel approach to speed up the adsorption energy prediction by applying Graph transformer model for predicting global minimum adsorption energy of adsorbate species on catalytic surface. Overall, the AdsGT approach is novel and has the potential to be explored further.


## Strengths:
Authors have demonstrated the capability of this new approach by benchmarking on three datasets curated after data-cleaning. The results are novel and are of interest to scientific community at large.

## Weakness:
As the reviewers highlighted, more work on comparison with other models would help researchers contextualize this work better.